# Comprehensive HRMS Chemical Characterization of Pomegranate-Based Antioxidant Drinks via a Newly Developed Suspect and Target Screening Workflow

**DOI:** 10.3390/molecules28134986

**Published:** 2023-06-25

**Authors:** Anthi Panara, Evagelos Gikas, Ilias Tzavellas, Nikolaos S. Thomaidis

**Affiliations:** Laboratory of Analytical Chemistry, Department of Chemistry, National and Kapodistrian University of Athens, Panepistimiopolis Zografou, 15771 Athens, Greece; panaranthi@chem.uoa.gr (A.P.); vgikas@chem.uoa.gr (E.G.); tzavell@chem.uoa.gr (I.T.)

**Keywords:** antioxidant drinks, novel workflows, HRMS, pomegranate, suspect screening methodology

## Abstract

Antioxidants play a significant role in human health, protecting against a variety of diseases. Therefore, the development of products with antioxidant activity is becoming increasingly prominent in the human lifestyle. New antioxidant drinks containing different percentages of pomegranate, blackberries, red grapes, and aronia have been designed, developed, and manufactured by a local industry. The comprehensive characterization of the drinks’ constituents has been deemed necessary to evaluate their bioactivity. Thus, LC-qTOFMS has been selected, due to its sensitivity and structure identification capability. Both data-dependent and -independent acquisition modes have been utilized. The data have been treated according to a novel, newly designed workflow based on MS-DIAL and MZmine for suspect, as well as target screening. The classical MS-DIAL workflow has been modified to perform suspect and target screening in an automatic way. Furthermore, a novel methodology based on a compiled bioactivity-driven suspect list was developed and expanded with combinatorial enumeration to include metabolism products of the highlighted metabolites. Compounds belonging to ontologies with possible antioxidant capacity have been identified, such as flavonoids, amino acids, and fatty acids, which could be beneficial to human health, revealing the importance of the produced drinks as well as the efficacy of the new in-house developed workflow.

## 1. Introduction

Pomegranate (*Punica granatum* L.), classified as a berry, is a member of the *Pinaceae* family and has been cultivated in the Mediterranean region (Turkey, Egypt, Tunisia, and Spain), as well as in India and Iran [1]. Pomegranate has been highly valued, due to its nutritional and medicinal properties, as well as its biological and free radical scavenging activities, which are attributed to the antioxidant phytochemicals derived from various parts of the plant (peel, seed, leaf, and flower) [2,3,4]. Pomegranate juice is a rich source of polyphenols, fructose, carbohydrates, glucose, and organic acids (i.e., ascorbic acid, citric acid, fumaric acid, and malic acid), while it contains several amino acids, including proline, methionine, and valine. Additionally, the presence of tannins and flavonoids, as the main type of polyphenols, indicates the pomegranate’s pharmacological potential, due to their antioxidant activity [1]. Ellagic acid, a metabolized form of ellagitannin is a powerful antioxidant, and it has an extensive applicability in plastic surgery, preserving the viability of skin flaps. Furthermore, anthocyanins (water-soluble pigments), flavan-3-ols, and flavanols are some of the flavonoids found in pomegranate related to plausible health benefits. Catechins, which can be found in both the juice and the peel of pomegranate, are vital to the biosynthesis of anthocyanins and have antioxidant and anti-inflammatory properties. It should be noted that all the flavonoids that appear in pomegranate have antioxidant capacity and contribute to the indirect suppression of inflammatory indicators, such as tumor necrosis factor-alpha (TNFα) [1]. It has been demonstrated that pomegranate fruits can be utilized for the treatment of human prostate cancer by inhibiting cell development and inducing apoptosis.

*Aronia melanocarpa*, commonly referred as “aronia,” is a *Rosaceae* family plant that is native to eastern North America and has lately also been grown in Europe [5]. Aronia extracts have demonstrated antioxidant, antidiabetic, and anti-inflammatory activity [6]. *Morus nigra*, which belongs to the *Moraceae* family, is also known as black mulberry [7], and is cultivated in India and China, as well in the Mediterranean region in Greece and Turkey [8]. A number of biological activities, including antidiabetic, antioxidant, anti-inflammatory, and antihyperlipidemic, have recently been linked to mulberry fruit [9]. *Vitis vinifera* L. (red grapes), a member of *Vitaceae* family [10], has traditionally been located from the South Caucasus toward the Mediterranean basin [11], whereas more recently new countries (United States, Australia, China) have been involved in the cultivation of the species [12]. Grapes are known to possess a wide array of biological activities such as antioxidant, antimicrobial, anti-inflammatory, and anti-cancer properties [13]. 

Pomegranate’s beneficial effect on human health has resulted in an increasing growth in its market share value. Therefore, the global pomegranate market is expected to be worth 248.4 million USD in 2022 and 338.6 million USD by 2028, with a CAGR of 5.3% during the review period [14]. The pomegranate-based juices in this study were created with the addition of secondary ingredients: blackberries, red grapes, and aronia. The selection of these ingredients, which could provide added value to the final drink, was based on their market availability, contribution to the antioxidant capacity of the juice, and effect on the flavor and inner flavor of the juice. There is an increasing trend to introduce these kinds of pomegranate-based drinks to the market. 

Regarding the evaluation of the drinks’ value and the determination of their antioxidant capacity, two trends appear in the literature. The first approach is related to the application of conventional techniques (i.e., DPPH, TEAC assay, Folin–Ciocalteu method) and the second one is linked to the determination of the antioxidant content of juice in terms of molecular species. The first approach is prone to various limitations, due to the nature of the matrix, with the most common being the interference from the background. To begin with, the DPPH (2,2-diphenyl-1-picrylhydrazyl) assay is a spectrometric method for determining antioxidants in solid or liquid matrices [15] via measuring substances’ ability to act as free radical scavengers or hydrogen donors. This method can be employed specifically for the estimation of the overall antioxidant capacity and the free radical scavenging activity of fruit and vegetable juices [16]. Furthermore, the Trolox Equivalent Antioxidant Capacity Assay (TEAC) measures spectrophotometrically the reduction of the radical cation ABTS+ (2,2-azinobis-(3-ethylbenzothiazoline-6-sulfonic acid) by antioxidant compounds [17], and it is commonly used in scientific research for analyzing foods’ and beverages’ antioxidant capacity [18]. Another frequently employed method for the determination of antioxidant capacity is the spectrophotometric method using the Folin–Ciocalteu reagent [19]. 

An alternative approach is the use of chromatography-based techniques for the determination of individual analytes that are connected to antioxidant activity. This approach ascertains the selectivity and credibility of the obtained results. High-pressure liquid chromatography with a photodiode array detector (HPLC-PDA) was used for the determination of phenolic acids in juices [2]. In order to further enhance the sensitivity and specificity of the acquired results, the recent trend is towards mass spectrometry-based methodologies, i.e., performing LC-MS/MS methods for the determination of organic acids [20], and HPLC/PDA/MS2 for the quantification of punicalagin, ellagic acids, and anthocyanidins [21]. A more comprehensive approach for obtaining a more holistic picture of the antioxidant landscape can be followed through the implementation of high-resolution (HR) techniques, i.e., conducting an analysis of anthocyanins and phenolic compounds via UHPLC-Orbitrap-MS [22,23], or an analysis of anthocyanins and other phenolic compounds (phenolic acids, ellagitannins, and flavonoids) in pomegranate juices via HPLC−DAD−ESI-qTOF-MS [24]. Additionally, an HRMS targeted and untargeted analysis in conjunction with chemometrics can be used alongside bioactive compound determination, as well as a reliable tool for pomegranate adulteration [25]. A wide array of various mass spectrometric analyzers has been employed for the quantitative determination of the mass spectrometry-based antioxidant capacity utilizing quadrupole based techniques (linear ion trap [26], and triple quadrupole based on MRM transitions [27]), as well as HRMS approaches, such as LC- qTOF-MS [28] and LC-orbitrap MS [29] for novel compound identification and structure annotation. Additionally, pomegranates’ metabolite profiling and the implementation of chemometrics has been conducted via nuclear magnetic resonance (NMR) spectroscopy [30].

The requirement for developing novel workflows capable of handling the massive amount of data derived from HRMS has emerged. Vendor-specific and open access software have been utilized to interpret the acquired data; however, the scientific community has noted the significance of employing and evaluating open-source software due to the variety of algorithms, the files’ compatibility between vendors, the codes’ transparency, the large community of software developers, and the capacity for their modification according to various licensing schemes. Nevertheless, there are still issues using non-targeted MS data derived either from data independent acquisition (DIA-bbCID) or data dependent acquisition (DDA-automs). DDA and DIA modes have been employed in conjunction, to maximize the benefits of each mode and ensure the mining of the largest features’ number. On the one hand, DDA is the most used strategy for compound elucidation, due to its cleaner and more easily interpretable spectra [31]. On the other hand, since DIA detects and fragments all ions in a sample, it empowers more thorough and repeatable analysis by collecting data within a wide range of known and unknown ions, while the fragmentation spectra are more complicated to interpret [32].

In this project innovative antioxidant pomegranate-based juices, that have been produced by the local vendor, were analyzed, which serves two supplementary purposes. The first aim is the molecular characterization of the drinks per se in terms of their quantitative and qualitative constitution. The second aim is the development of an advanced mass spectrometry novel workflow for the comprehensive characterization of the drinks in terms of the identification of compounds that demonstrate antioxidant activity. This has been accomplished via the compilation of an extensive suspect list of antioxidants for the characterization of bioactivity and the assembling of a literature-based suspect list for the usual comprehensive characterization of bioactivity. It is noteworthy that the aforementioned bioactivity-based characterization is a novel approach aiming towards a fast and efficient exploitation of the chemical domain. A new role for the combined suspect lists as a searchable database has been highlighted as having the potential for automated suspect screening and increasing the credibility of the identification results. Therefore, the bioactive-based and literature-based characterization of foods and beverages may pave the way for more comprehensive identification, while the utilization of open-source software provides an alternative yet efficient tool for the scientific community. 

## 2. Results

### 2.1. Suspect Screening of the Juice Employing Different Workflows—Qualitative Results

In total, 29 compounds were identified in all the investigated juices, employing the suspect screening methodology to reach different identification confidence levels depending on the available information. The levels of identification were based on the criteria set in the scientific work of Schymanski et al. [33]. Specifically, 17 compounds were identified at level 1, 10 compounds were identified at level 2a, one compound was identified at level 2b, and one compound was identified at level 3.

The identified compounds belonged to several categories with potential beneficial effects on human health. Specifically, eight organic acids (citric acid, malic acid, gallic acid, gentistic acid, chlorogenic acid, pyroglutamic acid, fumaric acid, and quinic acid), four fatty acids (linoleic acid, oleic acid, palmitic acid, and linolenic acid), three amino acids (phenylalanine, leucine, and norvaline) and one organic acid ester (ethyl gallate) were identified. Additionally, nine flavonoids and the metabolites thereof (ellagic acid, ellagic acid glucoside, quercetin, rutin, apigenin, 2-Phenylethyl beta-Dglucopyranoside, kaempferol, phlorizin, and verbacoside) and two sugars (fructose and glucosamine) were identified.

All the compounds identified are tabulated in Table 1, where the compound name, the molecular formula, the experimental and the predicted retention time, the theoretical and experimental *m*/*z* value of the precursor ion, and the ionization mode are provided. Additionally, the five most intense MS/MS fragments (if they existed) of the sample and their corresponding spectral data obtained either from the spectral library or the reference standards are presented. The cosine similarity scores of the investigated spectra for all samples, as acquired from MS-DIAL, are compared and the corresponding levels of identification are also presented. The samples are coded as 80%, 90%, and 100% based on the percentage of pomegranate, which is the basic ingredient.

### 2.2. Quantitative Analysis

The compounds, for which the analytical standards were available in the laboratory, were quantified using calibration curves of reference standards. Specifically, the concentrations of five organic acids (abscisic acid, chlorogenic acid, citric acid, gallic acid, and quinic acid), two flavonoids (quercetin and galangin), one flavonoid glucoside (verbascoside), and phenol glucoside (phlorizin) were determined. The compounds catechin, gentistic acid, epicatechin, genistein, p-coumaric, and pinobanksin were identified through target screening; however, their concentrations were below the limit of quantification (LOQ), defined as 0.5 mg/kg. The abovementioned concentrations with their corresponding standard deviation (SD) for the three investigated samples are presented. Additionally, the corresponding calibration equations in the form of y = (a ± S_a_)x + (b ± S_b_), as well as their determination coefficients are tabulated in Table 2. 

## 3. Discussion

### 3.1. Development of a Novel Workflow

The necessity for the development of a novel workflow that combines targeted and suspect screening with the existing DDA and DIA fragmentation methodologies has emerged recently. In the current data treatment software landscape, only MS-DIAL has the appropriate algorithms to perform both DDA and DIA (MS2Dec, CorrDec) analyses. On the other hand, MS-DIAL is not designed to perform target screening, whereas MZmine [34] is capable of performing target screening based on MS1 spectra and annotation based only on DDA fragmentation. Therefore, novel workflows that combined these two pieces of software were designed to overcome these issues.

The DDA and DIA approaches, producing the fragmentation of the molecular species, differ essentially to the precursor selection. Thus, in DDA, the precursor ion is selected, followed by fragmentation, whereas in DIA, no precursor ion is selected, instead, all ions are fragmented. Therefore, only DDA produces MS/MS spectra, while DIA generates MS/MS-like fragmentation but in MS1 spectra. Therefore, for the rest of the manuscript the term “fragmentation-derived spectra” will be used to describe the fragmentation pattern derived from either DDA (MS/MS) or DIA (high collision energy).

#### 3.1.1. Compilation of Suspect Lists

With the aim of deeply mining all the potential information relating to antioxidant contribution, a novel idea was conceived called bioactivity-driven interrogation. Therefore, focusing on a desirable property of the final product, substances of a specified activity were highlighted, bypassing the fuzzy information concerning the whole metabolic landscape. A new list that filtered the specific biological activity of the chemical space, i.e., emphasizing the antioxidant capacity (the antioxidant active compounds) in the current case, has been assembled. The compiled list has been entitled the “Bioactivity Driven Suspect List (BDSL)”.

Taking into consideration the metabolism of the most abundant antioxidant compounds of the BDSL, another list was compiled using combinatorial enumeration in order to predict products from potential metabolic pathways, such as glucolysation and methylation. This assembled list, called the “Virtual Metabolite Suspect List (VMSL)”, was generated using Smilib v2.0 [35,36], utilizing scaffolds, linkers, and building blocks according to the software. The scaffolds were the already identified natural products (NPs) of the BDSL, while the building blocks were one or two glucose units and/or a methyl group. Additionally, to retain the parent compounds, hydrogen was also selected as a building block. It should be noted that the metabolites of the compounds thereof could potentially have enhanced biological activity compared to their initial non-metabolized counterparts (i.e., quercetin may have a similar, or even enhanced, antioxidant capacity to quercetin glucosides, due to the latters’ different hydrophilicity). Alongside the aforementioned suspect lists, a literature-based list (LBL) has also been compiled. 

#### 3.1.2. MZmine-Based Workflow (MS1 Driven)

MZmine has a targeted feature detection module, which can interrogate the MS1 experiment using the information from a suspect list and construct the respective extracted ion chromatograms (EICs) as a feature list. These chromatographic peaks were prioritized based on the area under curve. It should be noted that the antioxidants need to be present in high quantities in order to exert their role. Thus, the most abundant antioxidants (MAA) were selected and used as input to construct a searchable database. This list will be utilized as a part of the MS-DIAL workflow. Therefore, these two lists also function in a confirmatory way as the simultaneous presence of a metabolite enhances the confidence. Thus, six common compounds were identified from both lists (VMSL and LBL): quercetin, kaempferol, apigenin, gentisic acid, gallic acid, and chlorogenic acid.

#### 3.1.3. MS-DIAL-Based Workflow (Fragmentation Driven)

MS-DIAL is deemed to be a valuable solution in order to exploit the DIA results, as well as DDA spectra. The compounds are annotated offline by comparing the DDA or DIA deconvoluted spectra to the corresponding ones from the samples. MS-DIAL searches against already assembled local libraries (i.e., the general list ESI (+/−)-MS/MS assembled from authentic standards, which is provided from the software’s download page). A novel idea has been conceived about the replacement of the abovementioned library with a narrowed version, encompassing only specific compounds of interest (i.e., a target/suspect version). In contrast to the untargeted mode, for which MS-DIAL has originally been used, this novel approach allows the software to function in the target/suspect mode. Therefore, this flexibility allows the construction of custom-made libraries, providing the capacity to narrow down the number of plausible candidates.

MSP files (editable with a simple text editor i.e., notepad, work pad etc.) consist of entries that include the candidates’ names, the molecular formula, the exact mass, the theoretical retention time, and the MS/MS fragments. Such files are publicly available from various sources, such as the GNPS, MS-DIAL, etc., webpages. These MSP files were adjusted to focus on the analytes of interest, thus serving as a database, which is essentially a suspect list. This approach offers the additional advantage of a more complete view for spectra comparison. Thus, the fragments included in this database correspond to experimental spectra and not biased/curated fragments as happens commonly in the compilation of suspect lists. These files are compatible with the MS-DIAL software, which offers the potential for processing both DDA and DIA data. Two lists have been generated (LBL and VMSL) and imported to MS-DIAL. The overall chemical space was searched with the aid of these two suspect lists, aiming to find the bioactive content in terms of antioxidants and to chemically characterize the final drinks in a comprehensive way.

### 3.2. The Antioxidant Activity of the Investigated Juices

Pomegranate, aronia, red grapes, and black berries, which are rich in antioxidants, were the ingredients in the juices generated. Pomegranate juice, which was the basic ingredient, is a rich source of antioxidants belonging to a variety of classes, including anthocyanins, ellagitannins, vitamin C [37], and citric acid [38]. Additionally, the ellagitannins’ metabolized by-products, known as urolithins, have potent antioxidant properties [37]. Pomegranate juice demonstrated the strongest antioxidant activity in comparison to other polyphenol-rich juices and drinks—such as apple, berry, concord grape, and orange juices, red wines, and iced tea—being nearly 20% higher than the abovementioned juices [18]. Aronia, which is added to the final product in a small percentage, has the highest antioxidant capacity among berries, as observed for aronia berries [39], in line with aronia juices, which ranked high, due to their polyphenol content [18]. The antioxidant capacity of the aronia plant is due to the presence of anthocyanins and cyanidin glycosides forms which mainly existed with glucoside moieties of 3-galactoside, 3-glucoside, 3-arabinoside, and 3-xyloside [40]. The antioxidant capacity in humans is linked to its action against the oxidation of red blood cells [41]. Grape juices, which participate in the final juice as a secondary ingredient, are very well-known antioxidant drinks, containing resveratrol, which is one of the most potent antioxidants and is found in grape skin and seeds. Furthermore, grapes contain high amounts of highly antioxidant substances, such as (+)-catechin, (−)-epicatechin, and procyanidins [42]. Grape juice has exhibited potent health benefits due to its antioxidant capacity, such as improved protection against blood LDL cholesterol oxidation [43], protecting against oxidative DNA damage, and inhibiting the production of oxidative damage products, such as 8-hydroxy-20 -deoxyguanosine (8-OHdG) [44]. Finally, blackberries, which are also used as a secondary ingredient in the juice produced, have a high antioxidant capacity, due to their high anthocyanin and ellagitannin content, as well as the presence of other phenolic compounds [45]. Blackberries additionally exhibit anti-inflammatory properties and thought to be a promising source of neuroprotective active compounds for age-related diseases due to their protective activity against oxidative damage [46,47]. 

### 3.3. Comparative Analysis of Antioxidant Juices

As the main aim of this endeavor was the development of a drink with enhanced antioxidant activity, various combinations of raw materials in different percentages (i.e., 3.3% and 6.6% from other juices) have been used. For clarification purposes, it should be noted that the secondary ingredients used were the juices of aronia, blackberries and red grapes. The contribution of the quantity for the selected antioxidants has been studied. Generally, three patterns have been observed based either on the targeted results or the corresponding peak areas (used for the substances for which reference standards were not available). An increasing trend of the investigated antioxidants when the percentage of other juices was higher (i.e., quercetin), showed either an opposite effect (i.e., ethyl gallate) or no effect (i.e., fructose). Thus, in the case of ethyl gallate, the drinks containing a higher percentage of the secondary ingredients contain lower amounts of this substance compared to the pure pomegranate drink. On the other hand, the amount of quercetin in juices (containing 80 and 90% pomegranate) is higher. Finally, the same amount of fructose has been determined in all three analyzed juices. This is depicted in Figure 1. No discrepancies in these patterns were noticed, which validates the results of the analysis.

The compounds quinic acid, kaempferol, quercetin, chlorogenic acid, rutin, and verbacoside were found to be higher in the juice supplemented with 6.66 % of each secondary raw material. Their elevated quantity is connected to the enhanced antioxidant activity.

### 3.4. Beneficial Role of the Identified Compound in Human Health

One prevalent criterion for the selection and the final percentage contribution of each ingredient in the produced drink is their antioxidant capacity in terms of the existence and content of bioactive substances. Hence, the presence of the antioxidants should be adequately high, as their activity is not excreted by the interaction of the substance with a pharmacological target/receptor. The obtained results revealed the presence of antioxidant compounds belonging to several categories (organic acids, fatty acids, amino acids, flavonoids, metabolites, etc.). The beneficial effect of the most important compounds identified is briefly discussed.

Ellagic acid is a well-known antioxidant that has been shown to be effective in preventing neurodegeneration by repairing mitochondrial damage and scavenging free radicals [48]. Quercetin is a powerful antioxidant known for its capacity to prevent tissue damage [49]. Kaempferol has anticarcinogenic, antioxidant, and anti-inflammatory [50], as well as antibacterial, antifungal, and antiprotozoal activities [51].

Fumaric acid has anti-inflammatory, neuroprotective, chemo preventive activities [52], and acts against multiple sclerosis (MS) [53]. Chlorogenic acid has antioxidant antibacterial, hepatoprotective, cardioprotective, anti-inflammatory, antipyretic, neuroprotective, anti-obesity, antiviral, antimicrobial, and antihypertension activity [54]. Citric acid is a secondary antioxidant [55], yielding synergistically to enhance primary antioxidants’ activity [56], while its chelating and acidulating properties are well-known [57]. 

## 4. Materials and Methods

### 4.1. Methodology for the Preparation of Pomegranate-Based Drinks

A thorough literature review was conducted for 16 potential raw materials that might serve as additional ingredients in the pomegranate-based juice to enhance its nutritional value. The raw materials investigated were *Prunus cerasifera*, *Vaccinium vitis-idaea* L., *Prunus cerasus*, *Aronia melanocarpa*, *Citrus*, *Ribes rubrum*, *Vitis vinifera* L., *Hippophae*, *Actinidia deliciosa*, *Opuntia ficus-Indica*, *Ficus carica*, *Rubus occidentalis*, and *Morus alba*, *Morus nigra.*

The *Morus nigra* (blackberries), *Aronia melanocarpa* (aronia), and *Vitis vinifera* L. (red grapes) were chosen based on their market availability, antioxidant contribution, and effect on the lingering flavor of the final product. 

The juices from aronia, blackberries, and red grapes were acquired from local small farmers in northern Greece and used without any further processing. The pomegranate fruits were collected during the October–November period from the region of North Greece. 

Then, two mixtures of juices were created, only differing in the proportion of their ingredients. Specifically, the percentage of pomegranate, blackberries, aronia, and red grapes were (90, 3.33, 3.33, 3.33, *v*/*v*) and (80, 6.66, 6.66, 6.66, *v*/*v*), respectively. Their flavor (sweetness, sour taste, acidity), lingering flavor (sour taste, acidity), as well as color, fragrance, and texture were assessed. The second mentioned juice had a higher overall score in the majority of the investigated categories.

The fruits were washed and, after selection, transferred to the appropriate apparatus to remove the peels and kernels. Afterwards, the juice was transferred into barrels in refrigerated conditions, and the next day, pasteurization and the hot filling procedure took place. The manufacturing process of the final product was initialized with the defrosting of the raw components (juices of pomegranate, red grape, blackberry, and aronia) until they reached room temperature. Afterwards, the transfer of the juices to tanks and their combination through stirring followed. Next, a pasteurization step in a tube heat exchanger at 83 °C was performed. Next, hot filling took place at 73 °C and the bottles were sealed using an automatic sealing machine. The juices’ temperature was decreased in a cooling tunnel. The bottles were kept at a temperature of 20 °C and protected from the light. The temperatures utilized for pasteurization and hot bottle filling during the manufacturing process are critical for the quality of the juice. The quality of the produced juice was ascertained using 83 °C for pasteurization and 73 °C for bottle filling as the optimum temperatures. Pasteurization temperature testing between 80 °C and 85 °C and bottle filling temperature testing between 73 °C and 75 °C was performed by the local industrial producer and the described optimized protocol was employed. 

### 4.2. Reagents and Materials

All the standards and reagents used were of analytical grade (<95%), unless explicitly stated. Methanol (MeOH, LC–MS grade) was purchased from Merck (Darmstadt, Germany), while formic acid 99% and acetic acid were acquired from Fluka (Buchs, Switzerland). Ammonium acetate and ammonium formate were obtained from Fisher Scientific (Geel, Belgium). The ultrapure water (H_2_O) was provided by a Milli-Q device (Millipore Direct-Q UV, Bedford, MA, USA). Regenerated cellulose syringe filters (RC filters, pore size 0.2 μm, diameter 15 mm) were acquired from Macherey-Nagel (Düren, Germany). Citric acid, chlorogenic acid, gallic acid, malic acid, quercetin, apigenin, phloridzin, L-phenylananine, leucine, fatty acid methylesters, D (-) fructose, catechin, epicatechin, pinobanksin, and p-coumaric acid were obtained from Sigma Aldrich (Stenheim, Germany. Verbascoside was purchased from HWI pharma services (Rülzheim, Germany), while quinic acid, genistein, gentistic acid, and galangin were acquired from Supelco (Stenheim, Germany). Apigenin was purchased from Alfa Aesar (Karlsruhe, Germany).

Stock solutions of the reference standards (1000 mg L^−1^) were prepared in MeOH (LC-MS grade) and stored at −20 °C in amber glass vials. A solution of 50 mg L^−1^ was prepared by the appropriate dilution of the individual stock standard solutions. Following that, dilutions with a mixture of MeOH: H_2_O (80:20, *v*/*v*) were performed in order to prepare working solutions with concentrations of 0.5, 1, 2.5, 5, and 10 mg L^−1^.

### 4.3. Sample Pre-Treatment for HRMS Analysis

In an eppendorf tube, 200 mg of the drink was weighed followed by the addition of 200 μL MeOH: H_2_O (80:20, *v*/*v*). The mixture was vortexed vigorously and filtered through RC syringe filters. The extracts were transferred to 2 mL autosampler glass vials and injected into the UPLC-QToF-MS system in both ionization modes.

### 4.4. Instrumentation

#### UPLC-QToF-MS Instrumentation

The chemical analysis of the pomegranate-based juice was carried out using ultra-high-pressure liquid chromatography-quadruple time of flight mass spectrometry (UPLC-QToF-MS) employed with an HPG-3400 pump (Dionex Ultimate 3000 RSLC, Thermo Fisher Scientific, Dreieich, Germany) coupled to a time-of-flight mass analyzer (Hybrid Quadrupole time of Flight Matic Bruker Daltonics, Bremen, Germany). Τhe chromatographic column utilized was an Acclaim RSLC 120 C18 column ((2.2 μm, 2.1 × 100 mm^2^) Thermo Fisher Scientific, Dreieich, Germany), equipped with a pre-column (Van guard Acquity UPLC BEH C18 (1.7 μm, 2.1 × 5 mm^2^, Waters, Ireland)) and its temperature (30 °C) was maintained during the analysis. In the positive ionization mode, the mobile phases consisted of (a) aq. 5 mM ammonium formate: MeOH (90:10, *v*/*v*) acidified with 0.01% formic acid and (b) 5 mM ammonium formate in MeOH acidified with 0.01% formic acid. In the negative ionization mode, the mobile phases were (a) aq. 10 mM ammonium acetate: MeOH (90:10, *v*/*v*) and (b) 10 mM ammonium acetate in MeOH. The same gradient elution program was used in both ionization modes. The gradient program is described in detail in a previous work by our group [58]. The values selected for the MS parameters were a capillary voltage of 3500 V, a nebulizer gas pressure of 2 bar (N_2_), a drying gas flow rate of 8 L min^−1^, and a capillary temperature of 200 °C. The sodium formate calibrant, which was prepared in H_2_O: isopropanol (50:50, *v*/*v*), was injected at the beginning of each run to calibrate the Q-ToF system on a daily basis.

According to the analytical method, the temperature of the LC column as well as the MS setting were optimized during a large series of experiments, as described in previous published works from our laboratory [59,60,61,62,63,64,65,66]. The chromatographic method used for the juice characterization is based on a generic protocol developed in our laboratory using more than 2000 substances, ascertaining the largest degree of separation. Furthermore, the mass spectrometric method has also been optimized in order to achieve the analytes’ highest ionization efficiency. This method is standardized to this kind of analysis for one additional reason, i.e., to ensure its compatibility with RTI methodology (http://rti.chem.uoa.gr/, accessed on 2 March 2023), which was also developed in our laboratory for suspect/ non-targeted analysis.

### 4.5. Mass Spectrometry Data Analysis

#### 4.5.1. Identification Confidence

The feature annotation was performed according to the Schymanski et al. scheme, considering the five levels of confidence in identifying a plausible candidate [33]. At level 5, the only confirmed information is the exact mass of interest, while there is no information concerning its molecular mass. At identification level 4, the candidate’s molecular formula is confirmed [67], whereas at the next level (level 3), a tentative identification via the evaluation of candidates’ MS/MS fragmentation is realized utilizing in silico fragmentation tools (MetFrag [68] or CFM-ID [69]). Additionally, at this identification level, prioritization methods, such as a retention time prediction [61] and ionization efficiency estimation [70], can be used to enhance the identification confidence. In cases in which diagnostic ions exist, the plausible candidate can reach identification level 2b. Potential candidates can reach identification level 2a when the corresponding MS/MS spectra are available at spectral libraries and their similarity score is higher than 0.7. At identification level 1, the candidates’ reference standards and their MS/MS spectra are available, the retention time being in accordance.

#### 4.5.2. Data Processing and Identification Workflows

##### Workflows for the Compilation of Suspect Lists

1.Bioactivity driven suspect list.

A suspect list compiled of 734 antioxidant substances was retrieved using Orange statistical language (version 3.33.0) through the text mining module using PubChem data. The molecular formula and the exact mass, alongside the compound name, were deposited in a csv file, which in turn was uploaded to MZmine 2.53. The raw data were calibrated and converted to mzxml files using Data Analysis software (Bruker Daltonics, Bremen, Germany) to be compatible with MZmine. The most abundant substances, i.e., those with the highest chromatographic peak areas, were selected for the evaluation of the drink’s antioxidant capacity. The mass spectral databases used for the assembling of the suspect list were: MoNa [71], MassBank-Europe [72], METLIN, Human Metabolome Database (HMDB) [73], and Global Natural Products Social Molecular Networking (GNPS) [74]. The features were annotated through the comparison of their MS/MS spectra with the corresponding ones from the spectral libraries or the reference standards in the cases where they were available in the laboratory. Due to the lack of entries concerning the metabolites derived from the enumeration process, as well as their MS/MS spectra from the aforementioned libraries, their fragmentation was estimated based solely on the characteristic diagnostic ions (i.e., for ellagic glucoside, the fragment of the aglucone part and the corresponding fragment of the sugar moiety). These pieces of information were also added to the VMSL list.

2.Comprehensive literature-based suspect list.

An exhaustive literature-based, text-mining-defined suspect list was created using the Orange statistical language. This list encompassed the compounds retrieved from PubChem that were specified for pomegranate, blackberries, red grapes, and aronia. Subsequently, a literature-based suspect list [2,20,21,22,23,24,25,30,75,76,77] was assembled in the traditional way and merged with the one obtained from the text mining procedure. This list was used for the suspect/target screening protocol and as a supporting tool to enhance the confidence of the acquired results from the BDSL. Furthermore, these two workflows acted synergistically to provide a holistic picture of the plant’s chemical composition. The workflows employed for the compilation of the suspect lists are illustrated in Figure 2. 

##### Methodology of the Development of Workflow MS1 Driven

The raw data were calibrated, converted to mzxml files, and uploaded to MZmine 2.53. The list of 734 antioxidants in csv form was imported and the items with the highest peak area derived from the feature list were used as scaffolds for the compilation of VMSL.

##### Methodology of the Development of Workflow MS2 Driven

The calibrated raw data were converted to abf files (ABF converter) [78], and then uploaded to the open-source MS-DIAL software (version 4.92) [79]. Both acquisition modes, DDA and DIA, were examined. DDA was selected for the most abundant compounds, whereas DIA was utilized for the compounds found in lower quantities. The different acquisition modes were processed separately. 

Based on the compounds mentioned in the literature (LBL), an in-house database was created and imported into MS-DIAL (MSP file format). Additionally, the VMSL (MSP file format) was also imported to MS-DIAL. These two MSP files were processed separately to evaluate the antioxidant content, as well as the compounds discovered through the comprehensive characterization of the drinks. The online “Retention time prediction tool” (available at http://rti.chem.uoa.gr/, last accessed 2 March 2023) was utilized to predict the theoretical retention time of each compound by uploading the canonical SMILES. For compounds with reference standards not available in the laboratory, the corresponding spectra were retrieved from public spectral libraries. A procedure blank was also prepared. The chromatographic peak areas of the procedure blank must be five-fold lower than the ones in the sample, in order to not be excluded as false positives.

The entire workflow is depicted in Figure 3.

#### 4.5.3. Target Screening Methodology

For the determination of the compound’s concentration, TASQ 1.4 (Bruker Daltonics, Bremen, Germany) was used. Quantification of the analytes was performed for the compounds with available reference standards which belong in the category of bioactive compounds, using standard based calibration curves according to a validated method developed in our laboratory [25]. Satisfactory linearity was achieved for all the analytes (R2 values ranging from 0.97 to 0.996). 

## 5. Conclusions

Pomegranate-based juices with antioxidant capacity have been designed, produced, and characterized employing novel suspect and target screening methodologies through open-source software using UPLC-QToF-MS. A total of 29 compounds, including fatty acids, amino acids, organic acids, and flavonoids and their metabolites were identified in the drinks via the developed methodologies in both ionization modes. The significant amount of quercetin, as well as the high concentration of citric acid, sparked a lot of interest, due to their plausible positive impact on human health.

In this context, novel suspect and target screening methodologies for the elucidation of drinks’ compounds have been developed to ascertain a faster and less effortful data treatment process, ensuring results with enhanced credibility. Bioactivity-/combinatorial- and literature-based lists have been assembled as a searchable database in combination with the mass spectrometry analysis using open-source software (MZmine, MS-DIAL). Furthermore, the manuscript poses the idea of compiling lists based on different activities besides antioxidant activity, as described in the context of this work. Therefore, screening plant material for other targeted bioactivities, such as anticancer, antibiotic, antidiabetic activities, etc. is an appealing approach proposed in the framework of this research. It is noteworthy that the assembled lists are literature-based and, therefore, not dependent on the availability of reference standard compounds in the respective laboratories, giving the opportunity to explore the existence of related activity compounds. Extrapolating this idea, a similar activity-based approach can be applied to different matrices and different activities, e.g., the toxic activity of biological samples to identify the sources of maladies.

## Figures and Tables

**Figure 1 molecules-28-04986-f001:**
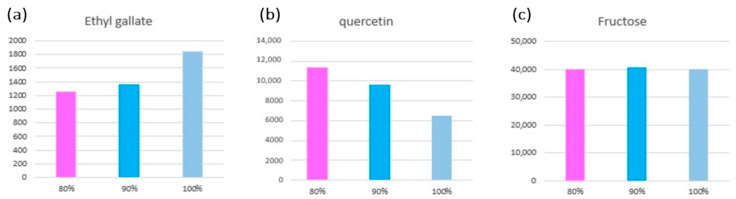
Quantity of (**a**) ethyl gallate, (**b**) quercetin, and (**c**) fructose for the designed juices with 80%, 90%, and 100% pomegranate.

**Figure 2 molecules-28-04986-f002:**
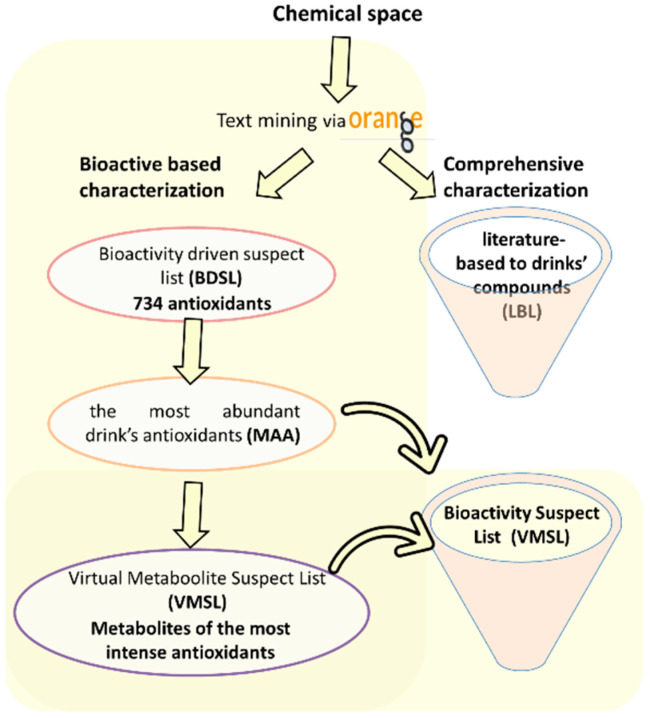
Workflow of suspect lists compilation.

**Figure 3 molecules-28-04986-f003:**
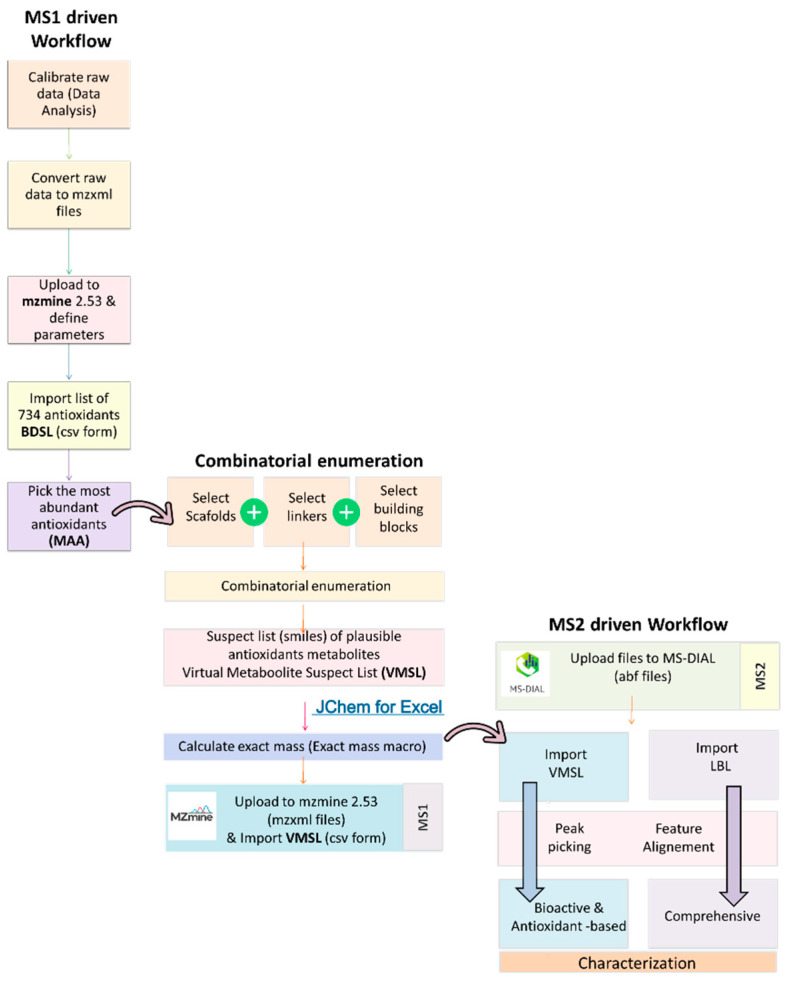
Workflow of the novel suspect screening methodology.

**Table 1 molecules-28-04986-t001:** Compounds identified through suspect screening in the investigated juices.

Compound Name	Chemical Formula	Exp. t_R_ (min) (Reference Standard) ^a^	Pred. t_R_ (min)	Application Domain ^b^	Exp. *m*/*z* ^c^	Theor. *m*/*z*	ESI Mode	MS/MS Explained Fragments ^d^	Reference MS/MS Spectra ^e^	Total Score			Level of Identification/Database Reference ^i^
										80% ^f^	90% ^g^	100% ^h^	
Citric acid	C₆H₈O₇	1.2 (1.2) ^a^	1.1	Box 1	191.0213	191.0197	−ESI	57.0353	57.0354	0.76	0.75	0.76	1
								87.0092	87.0089				
								111.0094	111.0088				
								191.0198	191.0199				
Malic acid	C_4_H_6_O_5_	1.2 (1.1) ^a^	1.0	Box 1	133.0131	133.0143	−ESI	71.0144	71.0139	0.69	0.72	0.65	1
								115.0047	115.0022				
								133.0127	133.0138				
Fructose	C_6_H_12_O_6_	1.4 (1.3) ^a^	1.7	Box 1	179.05754	179.0561	−ESI	89.0224	89.0246	0.66	0.64	0.69	1
								179.0560	179.0558				
Gallic acid	C_7_H_6_O_5_	1.6 (1.5) ^a^	2.9	Box 2	169.0151	169.0142	−ESI	69.0354	69.0346	0.66	0.66	0.76	1
								97.0271	97.0295				
								125.0245	125.0244				
Gentisic acid	C_7_H_6_O_4_	2.2 (2.4) ^a^	3.0	Box 1	153.0199	153.0193	−ESI	108.0191	108.0217	0.70	0.70	0.76	1
								109.0289	109.0295				
Chlorogenic acid	C_16_H_18_O_9_	2.9 (2.9) ^a^	3.5	Box 1	353.0896	353.0878	−ESI	161.0264	161.0233	0.74	0.76	0.76	1
								173.0479	173.0447				
								191.0557	191.0555				
								192.0564	192.0589				
Fumaric acid	C_4_H_4_O_4_	1.3	1.9	Box 1	115.0035	115.0037	−ESI	71.0136	71.0136	0.68	0.71	0.74	2a MzCloud no 1274
								72.9928	72.9925				
								115.0035	115.0040				
Quinic acid	C_7_H_12_O_6_	1.3 (1.3) ^a^	1.2	Box 1	191.0564	191.0561	−ESI	85.0291	85.0299	0.62	0.64	0.69	1
								191.0564	191.0558				
Phenylalanine	C_9_H_11_NO_2_	3.0	4.2	Box 2	164.0726	164.0717	−ESI	72.0091	72.0099	0.63	0.66	0.61	1
								147.0447	147.0448				
								164.0728	164.0712				
Leucine	C_6_H_13_NO_2_	2.8	1.8	Box 1	132.1020	132.1019	+ESI	58.0654	58.0634	0.69	0.68	0.69	1
								69.0697	69.0686				
								86.0964	86.0956				
								87.0989	87.0987				
								132.1019	132.1013				
Norvaline	C_5_H_11_NO_2_	4.5	1.2	Box 3	118.0646	118.0863	+ESI	65.03932	65.036	0.71	0.73	0.72	2a MonA ID: FiehnHILIC002191
								117.0584	117.057				
								118.0658	118.062				
Quercetin	C_15_H_10_O_7_	7.3 (7.3) ^a^	7.0	Box 1	301.0361	301.0354	−ESI	151.0042	151.0037	0.68	0.61	0.70	1
								169.0170	169.0135				
								179.0004	178.9996				
Rutin	C_27_H_30_O_16_	5.7 (5.7) ^a^	6.2	Box 1	609.1472	609.1461	−ESI	300.028	300.0256	0.62	0.63	0.68	1
								301.0312	301.0366				
Apigenin	C_15_H_10_O_5_	8.3 (7.9) ^a^	7.6	Box 1	269.0460	269.0455	−ESI	117.0359	117.0341	0.61	0.66	0.61	1
								151.0077	151.0029				
								269.0463	269.0459				
Kaempferol	C_15_H_10_O_6_	7.6	7.2	Box 1	285.042	285.0405	−ESI	133.0305	133.0297	0.76	0.78	0.71	2a GNPS ID: VF-NPL-QEHF014174
								151.0030	151.0039				
								175.0386	175.0388				
								285.0421	285.0400				
Verbascoside	C_29_H_36_O_15_	5.0 (4.8) ^a^	8.4	Box 4	623.1993	623.1981	−ESI	161.0319	161.0244	0.62	0.69	0.68	1
								162.0263	162.0278				
Phloridzin	C_21_H_24_O_10_	5.9 (5.8) ^a^	8.2	Box 2	435.1291	435.1297	−ESI	167.0362	167.0340	0.66	0.67	0.75	1
Ethyl gallate	C_9_H_10_O_5_	4.9	5.0	Box 1	197.0473	197.0455	−ESI	123.0061	123.0086	0.62	0.63	0.62	3 FoodB ID:FDB012004
								140.0102	140.0118				
								168.0066	168.0074				
								169.0149	169.0146				
								197.0460	197.0460				
Linoleic acid	C_18_H_32_O_2_	13.5 (13.5) ^a^	12.9	Box 1	279.2336	279.2330	−ESI	279.2327	279.2328	0.61	0.60	0.70	1
								280.2345	280.2333				
Oleic acid	C_18_H_34_O_2_	14.0 (14.0) ^a^	13.3	Box 1	281.2486	281.2486	−ESI	281.2484	281.2468	0.69	069	0.74	1
								282.2526	282.2508				
Palmitic acid	C_16_H_32_O_2_	13.8 (13.8) ^a^	13.0	Box 1	255.2334	255.2330	−ESI	255.2329	255.2327	0.84	0.83	0.86	1
								256.2366	256.2364				
								257.2349	257.2395				
Linolenic acid	C_18_H_30_O_2_	13.0	12.4	Box 1	277.2170	277.2173	−ESI	277.2198	277.2180	0.91	0.92	0.95	2a MoNA ID: MetaboBASE0976
Ellagic acid	C_14_H_6_O_8_	4.6	4.7	Box 1	300.9993	300.9990	−ESI	201.0183	201.0200	0.77	0.74	0.80	2a MoNA ID:FiehnHILIC001170
								229.0143	229.0144				
								283.9960	283.9950				
								299.9924	299.9900				
								300.9992	300.9994				
Glucosamine *	C_6_H_13_NO_5_	1.8	1.4	Box1	162.0764	162.0760	+ESI	60.0450	60.0443	0.94	0.97	0.93	2a GNPS ID: CCMSLIB00005464276
								72.0450	72.0435				
								84.0450	84.0445				
								85.0290	85.0284				
								162.0760	162.0744				
2-Phenylethyl beta-D-glucopyranoside *	C_14_H_20_O_6_	6.5	6.2	Box 1	302.1616	302.1600	+ESI	81.0337	81.0330	0.89	0.92	0.94	2a GNPS ID: CCMSLIB00000854907
								85.0287	85.0270				
								97.0289	97.0280				
								105.0707	105.0710				
								127.0340	127.0400				
Pyroglutamic acid	C_5_H_7_NO_3_	2.4	2.1	Box1	130.0511	130.0507	+ESI	84.0455	84.0460	0.94	0.93	0.91	2a MassBank ID: PR311148
								85.0483	85.0450				
								129.0190	129.0220				
								130.0508	130.0510				
4-Hydroxyquinoline	C_9_H_7_NO	4.5	6.1	Box 2	146.0606	146.0600	+ESI	77.0389	77.03700	0.93	0.97	0.99	2a RIKEN PLaSMA ID: RIKENPlaSMA000824
								91.0541	91.0560				
								101.0395	101.0440				
								146.0598	146.0610				
Dihydrozeatin	C_10_H_15_N_5_O	5.8	4.31	Box 2	222.1351	222.1349	+ESI	69.0699	69.0710	0.91	0.88	0.92	2a MoNA ID: FiehnHILIC000308
								136.0615	136.0620				
								148.0626	148.0620				
								204.1227	204.1250				
								222.1347	222.1349				
Ellagic acid glucoside	C_20_H_16_O_13_	3.7	4.5	Box 1	463.0514	463.0518	−ESI	300.9976					2b diagnostic ion Using Smilib

^a^ retention time of the reference standard; ^b^ Development and Prediction of Retention Time Indices, available at http://rti.chem.uoa.gr (accessed on 02/03/2023); ^c^ experimental *m/z* value with error ± 0.005 Da, [M + H]^+^ for +ESI and [M − H]^−^ for −ESI (* with the exception of glucosamine, whose *m/z* value corresponds to [M-H_2_O + H]^+^, and 2-phenylethyl beta-D-glucopyranoside whose *m/z* value corresponds to [M + NH_4_]^+^ ); ^d^ top-five most intense explained peaks (if they existed); ^e^ MS/ MS fragments of the spectra reference standard or mass spectral library; ^f^ 80% pomegranate; ^g^ 90% pomegranate; ^h^ 100% pomegranate; ^i^ for identification level 2a the database entry is referenced.

**Table 2 molecules-28-04986-t002:** Quantification results.

Analyte	Concentration (mg/kg) ± SD (*n* = 3)	Concentration (mg/kg) ± SD (*n* = 3)	Concentration (mg/kg) ± SD (*n* = 3)	Equation of the External Calibration Curve y = (a ± Sa)x + (b ± Sb)	Determination Coefficient R^2^
	80% ^a^	90% ^b^	100% ^c^		
Abscisic acid	0.28 ± 0.02	<LOQ	<LOQ	y = (29,407± 1108)x + (13,932 ± 5655)	0.994
Chlorogenic acid	4.07 ± 0.33	1.35 ± 0.08	0.52 ± 0.05	y = (299,437 ± 17,948)x + (236,799 ± 91,609)	0.98
Citric acid	203 ± 18.9	204 ± 21.2	199 ± 18.4	y = (114,212 ± 3871)x − (27,802 ± 19,759)	0.991
Galangin	1.41 ± 0.86	0.65 ± 0.05	<LOQ	y = (179,124 ± 11,301)x − (9729 ± 57,678)	0.98
Gallic acid	5.72 ± 0.41	4.57 ± 0.41	5.11 ± 0.47	y = (46,623 ± 3535)x + (57,556 ± 18,043)	0.98
Phloridzin	1.01 ± 0.09	0.65 ± 0.05	0.65 ± 0.06	y = (304,319 ± 25,920)x + (287,534 ± 132,294)	0.97
Quinic acid	0.75 ± 0.09	0.38 ± 0.04	<LOQ	y = (121,371 ± 1970)x + (431,844 ± 10,055)	0.993
Verbascoside	0.87 ± 0.12	0.51 ± 0.05	<LOQ	y = (58,119 ± 1053)x − (14,120 ± 5376)	0.996
Quercetin	13.1 ± 0.45	11.6 ± 0.56	11 ± 0.48	y = (71,193 ± 4859)x + (51,014 ± 24,802)	0.992

^a^ 80% pomegranate; ^b^ 90% pomegranate; ^c^ 100% pomegranate.

## Data Availability

Data sharing not applicable.

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
