# Peer review of "Comprehensive HRMS Chemical Characterization of Pomegranate-Based Antioxidant Drinks via a Newly Developed Suspect and Target Screening Workflow"

_molecules, 2023, doi:10.3390/molecules28134986_

Round 1
Reviewer 1 Report
The manuscript titled “Comprehensive HRMS chemical characterization of pomegranate-based antioxidant drinks via a newly developed suspect and target screening workflow” describes the development and application of a new HRMS method to explore the antioxidant drinks media in an automatic target screening way. The manuscript is correctly written, scientifically sound, and clearly presented in good written English. Presented manuscript deals with the design and control of the composition of functional food, which is a very current topic with a great possibility of application in the food industry.
Major Concern:
Please explain precisely whether ethyl gallate is present in the examined energy drinks as a natural part of pomegranate juice or other juices that are added to energy drinks or is added to energy drinks separately as a special additional ingredient, as a synthetic antioxidant (food additive).
If ethyl gallate is the natural ingredient of pomegranate juice or other natural juices, was its proportion in the natural juices determined before pomegranate juice or other fruit juices were added to the energy drinks? What is the natural level/concentration of ethyl gallate in these specific natural juices?
Minor concerns:
In the Introduction please elaborate in more detail on the capabilities of MS methods in antioxidant compound determination in foods samples. Also, explain in a simple manner possibilities of the use of workflow for data analysis in HRMS since the board possible reading the audience of this article.
Please give more details about the temperature programs and how you arrived at the final optimized method.
Moderate editing of the English language is required.
Author Response
Please find attached the responses to reviewer 1.

Reviewer 2 Report
The manuscript by Nikolaos S. Thomaidis et al. is well written and well researched. I particularly find the concept interesting and up-to-date with direct application to food industry. However, it lacks of scientific direction. Specifically, it is not clear if the authors present a novel suspect screening workflow since lots of similar works exist or the work is focused on the exploration of the enhanced antioxidant ingredients of the juices. It is my suggestion to add for example a section concerning the antioxidant activity of the juices under investigation. Also, many details concerning the juices' origin and preparation are missing. A comparison with similarly enhanced antioxidant juices that are already distributed at markets would also be stimulating. Finally, on my opinion, section 3.3 does not offer significant data and should be completely omitted or incorporated in the 2.1 section. In the Table 1 some experimental m/z values are missing, e.g., for rutin.
Generally, I recommend this manuscript to be published in Molecules but after major revisions.
Minor editing
Author Response
Please find attached the responses to reviewer 2.

Round 2
Reviewer 2 Report
The revised manuscript is accepted as it is.
Minor editing of english language
Author Response
Thanks for your peer review check